# Mechanisms of E-Cigarette Vape-Induced Epithelial Cell Damage

**DOI:** 10.3390/cells12212552

**Published:** 2023-10-31

**Authors:** Emily Auschwitz, Jasmine Almeda, Claudia D. Andl

**Affiliations:** Burnett School of Biomedical Sciences, University of Central Florida, Orlando, FL 32816, USA

**Keywords:** e-cigarette vape, inflammation, DNA damage, reactive oxygen species, cell signaling, cancer

## Abstract

E-cigarette use has been reported to affect cell viability, induce DNA damage, and modulate an inflammatory response resulting in negative health consequences. Most studies focus on oral and lung disease associated with e-cigarette use. However, tissue damage can be found in the cardio-vascular system and even the bladder. While the levels of carcinogenic compounds found in e-cigarette aerosols are lower than those in conventional cigarette smoke, the toxicants generated by the heat of the vaping device may include probable human carcinogens. Furthermore, nicotine, although not a carcinogen, can be metabolized to nitrosamines. Nitrosamines are known carcinogens and have been shown to be present in the saliva of e-cig users, demonstrating the health risk of e-cigarette vaping. E-cig vape can induce DNA adducts, promoting oxidative stress and DNA damage and NF-kB-driven inflammation. Together, these processes increase the transcription of pro-inflammatory cytokines. This creates a microenvironment thought to play a key role in tumorigenesis, although it is too early to know the long-term effects of vaping. This review considers different aspects of e-cigarette-induced cellular changes, including the generation of reactive oxygen species, DNA damage, DNA repair, inflammation, and the possible tumorigenic effects.

## 1. Introduction to Electronic Cigarettes or Electronic Nicotine Delivery Devices (ENDS)

E-cigs were initially marketed as a smoking cessation aid for addicted adults, but the promise of a safer product and flavored cartridges heavily attracted adolescent users [1]. Product sales initially doubled nearly every year, and continue to show an annual increase in usage, particularly among the underage demographics. E-cigs are primarily composed of a solution cartridge, a vaporization chamber, a coil (the heating element), and a battery [2]. Users operate the device by inhaling from the mouthpiece after the liquid contained in the cartridge is heated. The heated liquid is vaporized by the coil to be ingested into the oral cavity and delivered to the respiratory tract [2]. The devices come in various shapes and sizes, which can alter the vaping experience for the user.

Advocacy groups such as the Smoke Free Alternatives Trade Association (SFATA), the Consumer Advocates for Smoke-free Alternatives Association (CASAA), the American Vaping Association (AVA), and others lobby against anti-vaping legislation and seek to promote the usage of e-cig products for adults looking for safer alternatives to traditional cigarettes [3]. The use of e-cigs as a smoking cessation tool for adult smokers is a major argument of vaping supporters [4]. In 2018, a study revealed that 15% of smokers were able to successfully quit smoking by exclusively using e-cigarettes. From this study, only 3% of smokers were able to quit smoking using a nicotine replacement therapy, such as a nicotine patch or nicotine gum. About 6% were able to quit smoking without any tobacco alternatives [5]. While e-cigarettes are promoted as smoking cessation tools for adults, they attract mostly young users: Only 6.7% of adults are e-cig users while 28.4% are young adults (18–24 years of age) with 19.6% still attending high school [6,7,8]. The U.S. Surgeon General stated [8] that “we have never seen use of any substance by America’s young people rise as rapidly as e-cigarette use”, which presents a new challenge. In 2019, nearly 30% of high school students reported use of a vaping device [9]. A major appeal of e-cigs to adolescents is the variety of flavors in which the products are available [9,10]. According to most reports, fruit and candy flavors are the most popular among teens. A flavor ban legislation was created for the purpose of lessening some of the appeal of e-cigs to underage users [9,10]. Unfortunately, most of these efforts were futile as flavor bans were either short-term or legally challenged. The flavors found in e-cig solutions contain compounds such as aldehydes, benzyl alcohol, terpenes, pyrazines, menthol, menthone, and ethyl maltol [11]. While these compounds are frequently used for food flavoring and cosmetic scents, their toxicity when inhaled into the lungs is unknown [12]. 

In this review, we will highlight the risks associated with e-cig vaping with a specific focus on lung and oral epithelial cells. 

## 2. Mechanisms of E-Cigarette-Induced Health Effects

### 2.1. The Role of E-Cigarette Compounds in Disease Initiation

E-cigarettes are widely promoted as safer alternatives to traditional smoking. Over 4000 harmful and potentially harmful constituents (HPHCs) have been identified in traditional cigarette smoke [13]. Cigarette smoke has two phases: the particulate phase and the gaseous phase [14]. In the particulate phase, the primary toxicants are nicotine, tar, polynuclear aromatic hydrocarbons (PAHs), and tumor-stimulating substances such as indole and carbazole [15]. Although the gaseous phase mainly comprises nitrogen, oxygen, and carbon dioxide, the following HPHCs have also been found in the gaseous phase of cigarette smoke: hydrocyanic acid, hydrazine, ciliotoxins, acetaldehyde, ammonia, acrolein, formaldehyde, and carbon monoxide. Comparatively, the HPHCs found in e-cigarette smoke are anywhere between 9 and 450 times lower than what is found in traditional cigarettes [15]. 

Aldehyde components such as acetyl aldehyde, formaldehyde, and acrolein are amongst the most harmful in tobacco smoke and also in e-cig vape. Interestingly, aldehydes such as formaldehyde, which cause reactive oxygen species and inflammation in the lung [16,17], have been detected in both flavored and unflavored e-cig liquids regardless of nicotine presence [16]. The heating of the vape device coil causes thermal degradation, oxidation, and pyrolysis of even non-toxic compounds such as the carrier liquids propylene glycol and glycerol [18,19,20]. Dripping is a vaping method in which a vape user manually adds a few drops of vapor to the atomizer. Dripping and/or dry puff were suggested to result in particularly high levels of aldehyde [21,22]. Aldehyde at high concentrations can be dangerous, but the concentration of the flavors in the lungs while vaping is not really known [17]. Heating a mixture of aldehydes, propylene glycol (PG), and vegetable glycerol (VG) creates toxic byproducts, which lead to respiratory damage and can also have possible carcinogenic affects [12,23]. 

Reports about lung disease in association with butter-flavored popcorn have been published, e.g., the so-called popcorn lung [24], highlighting the potential toxicity of buttery flavors. Diacetyl together with acetoin are the key ingredients responsible for the buttery flavor in e-cig liquids [25]. These and other volatile compounds have been shown to induce DNA damage and are linked to lung cancer (see Section 2.4 regarding DNA damage and Section 2.5 for tumorigenesis). Putting the vape user further at risk is the fact that the concentration of these compounds is not regulated in e-cigs [12]. 

The toxic combination of substances found in e-cigs leaves today’s youth e-cig users at a particular risk for adverse health effects. Recent studies also indicate advertisement from the vaping industries may encourage usage by young adults [26]. Another study implicates social media as a primary source influencing the use of vape products among teens and young adults [27]. In addition to the increasing use of e-cigs among adolescents, there is concern about how nicotine impacts the developing brain of young adults. Brain development starts embryonically and continues into early adulthood. Individuals are nuanced by the synaptic connections of their neurons, which are mass produced during childhood. During adolescence, synapses undergo the process of synaptic pruning. This modification eliminates unnecessary neuronal connections, and its completion produces a matured adult brain. Synaptic pruning is impacted by both environmental and genetic factors, and impeding this process may result in maladaptive behaviors in adulthood. Nicotine is the primary psychoactive and addictive component of tobacco, although it is not the sole source of harm that stems from tobacco use [28]. It acts on nicotinic acetylcholine receptors found in the brain and peripheral nervous system [28]. Although accompanied by negative side effects, nicotine activates the reward systems in the brain and increases the likelihood of continued usage. Overtime, nicotine induces changes in the neuronal circuitry, which alters the sensitivity of receptors to the drug and greater doses of nicotine are needed to produce the same rewarding effects for the user [29]. Some studies show adolescents have increased sensitivity to the addictive effects of nicotine because their neurological reward systems are not yet fully matured [29,30]. Additionally, nicotine use has been associated with impaired memory and cognitive function in teens, although these studies have been flagged with significant limitations that may impact the validity of the results [29,30]. According to e-cig manufacturers, a single e-cigarette device may contain as much nicotine as a pack of 20 conventional cigarettes [30]. A recent CDC study demonstrated that 99% of the e-cigarettes sold in the U.S. contain nicotine, some labels do not disclose whether they contain nicotine, and some of them contain nicotine even though they are marketed as 0% nicotine [31]. The overall nicotine concentration in Juul pods (59 mg/mL for 5% or 35 mg/mL for 3%) and similar devices is higher than that in traditional cigarettes, averaging to an equivalent of the amount of nicotine in a pack of cigarettes [32]. Modifications such as nicotine salts were introduced to enhance the sensory quality of e-cig vapes. The promised appeal lies in a “less harsh vape allowing for better tolerance of higher nicotine salt e-juice” (brand website https://www.elementvape.com/). Nicotine salts may have a stronger inflammatory effect on the lung epithelium than regular nicotine-containing e-liquids [33]. Also, the nicotine found in e-cigarettes still metabolizes to N’-nitrosonornicotine (NNN), which is a highly carcinogenic nitrosamine compound [34]; see Section 2.4.

Smoking and vaping have also been cited as gateway behaviors to other forms of substance abuse. The use of tobacco products is 5 times more common among groups where other substance abuse is occurring [28]. Because nicotine and alcohol are legal in the U.S., these are often the first types of substance exposure individuals will experience [29]. Nicotine has also been linked to increased sensitivity to cocaine [35]. In 2011, a study demonstrated that nicotine exposure resulted in the increased expression of the *FosB* gene, which is known to be associated with cocaine addiction in the striatum of the brain [35]. Consequently, the FDA released a statement on the safety of smoking mediums in 2022, stating that no tobacco products are safe for consumption, including e-cigarettes [36].

### 2.2. E-Cig-Modulated Inflammatory Signaling

Inflammation is part of the normal host immune response to defend against invading pathogens, cellular damage, and other noxious stimulants [37]. The process of inflammation involves a highly regulated cascade of molecular events, which is usually short-lived. If inflammatory signaling is activated long-term and becomes chronic, the effects are damaging [37] and even associated with tumorigenesis [38].

Pattern-recognition receptors on host cells recognize pathogen-associated molecular patterns (PAMPs) and danger-associated molecular patterns (DAMPs), and this event induces inflammation [37]. In humans, the pattern-recognition receptors associated with inflammation are the family of Toll-like receptors, the IL1 receptor, the IL6 receptor, and the TNF receptor [37] (Figure 1). Damaged cells release chemokines and other factors to attract pro-inflammatory immune cells: Neutrophils arrive first, followed by macrophages, lymphocytes, and mast cells. The functions of macrophages are antigen presentation, cytokine production, and phagocytosis. Mast cells effectively carry out inflammation responses by releasing other pro-inflammatory molecules. Once the damage has been successfully resolved, the pro-inflammatory cells are no longer recruited to the site and the inflammation resolves [37]. 

If inflammation remains unresolved or is inappropriately stimulated, it can promote carcinogenesis by inciting DNA damage and oxidative stress, as well as via inhibition of certain aspects of immune cell activity [39]. Both traditional cigarettes and e-cigs have the capacity to incite inflammation. Pro-inflammatory molecules found in the tar phase of cigarette smoke cause inflammation in exposed tissues [40]. Traditional cigarettes lead to chronic inflammation in the gums by promoting pathogenic bacterial growth; this can lead to the development of periodontitis [41]. Similarly, e-cigs have been directly linked to adverse effects on periodontal health: Clinical attachment loss, a combination of gingival recession and the formation of gum pockets, causes the tooth to loosen and eventually fall out. A clinical study enrolling 101 subjects showed that e-cig users and smokers had markedly higher rates of severe periodontal disease than non-smokers: Clinical attachment loss was measured at 2.8 mm for e-cig users and 3.5 mm for smokers compared to 2.2 mm for non-smokers. Interestingly, in e-cig vapers, clinical attachment loss was significantly worse at 3.1 mm at 6-month follow-up compared to conventional smokers at 3.4 mm [42]. Notably, the oral microbiome changes in response to vape as it influences the growth patterns of commensal and pathogenic bacteria. We and others have shown that e-cig vape supports the growth of periodontal pathogens. Assessing 101 periodontal patients, Xu et al. identified an altered oral microbiome with increased members of Filifactor, Treponema, and Fusobacteria [43]. The enrichment of these taxa correlated with significantly increased levels of the pro-inflammatory cytokines TNFα, IL6, and IL1B among e-cig users, Figure 1. In contrast, the cytokine IL4 was lower among e-cig users than non-users; IL4 tends to be reduced in people with gum disease and increases after treatment, which suggests that certain bacteria in the mouths of e-cig users may be actively suppressing immune responses [43,44]. We have shown that e-cig aerosols inhibit the growth of the oral residents *Streptococcus sanguinis* and *gordonii* but not that of the cariogenic *S. mutans*. E-cig aerosols further stimulated *S. mutans* biofilm formation, supporting oral colonization (Figure 1). Our study suggests that e-cig aerosols have the potential to dysregulate oral bacterial homeostasis by suppressing the growth of commensals while promoting colonization of the opportunistic pathogen *S. mutans* [45]. As a cariogenic bacterium, *S. mutans* also favors growth conditions high in carbohydrates as it can metabolize the most common dietary sugars (fructose, glucose, and sucrose) to lactic acid [46]. Sweet vape flavors contain sugars such as sucrose, fructose, and glucose [47] as depicted in Figure 1, therefore promoting *S. mutans* growth and colonization, causing a rise in lactic acid, which is the root cause of caries [48].

Our studies have further shown that e-cig aerosols decrease the secretion of cytokines in oral epithelial cells immediately following exposure, leaving the vape user at potential risk for bacterial colonization and infection [49]. We have shown that *Staphylococcus aureus,* a pathogen found to be enriched in oral tumors, colonizes the oral epithelium after e-cig aerosol exposure and, over time, induces an enhanced production of pro-inflammatory proteins such as cycloxygenase-2 (COX2), which is further elevated in response to enhanced bacterial adhesion by S. aureus in the oral epithelium [49]. Sundar et al. demonstrated that increased levels of IL8, the inflammatory mediator prostaglandin-E2, and COX2 are associated with upregulation of the receptor for advanced glycation end products, or RAGE, by e-cig exposure-mediated carbonyl stress in the gingival epithelium [50]. Their studies also show the subsequent induction of DNA damage (see Section 2.4).

Lung damage is one of the primary pathologies associated with the use of tobacco products. The smoking of traditional cigarettes, the most common cause of lung cancer, can also lead to the development of chronic obstructive pulmonary disease (COPD), and may worsen existing asthma conditions [51]. Similar as described for the oral cavity, smoking can also lead to the development of bacterial and viral infections by perturbing the normal immune functions of the pulmonary system [52]. Traditional cigarette smoke exposure affects the function of the mucociliary escalator by inhibiting ciliary beating and increasing mucus secretion, which can trap bacteria and viruses leading to infections [52]. Traditional cigarette smoking impairs the pulmonary immune response by manipulating a variety of key immune cells such as neutrophils, natural killer (NK) cells, and alveolar macrophages [53]. Studies show that acrolein, found in traditional cigarette smoke and e-cig vape, reduces the phagocytic capacity of neutrophils by suppressing the Fc receptors on the cells [53], see Figure 1. NK cells kill infected host cells and suppress tumor formation by releasing the cytotoxic contents of their granules and promoting apoptosis [52]. Studies using mice revealed decreased tumor-clearing abilities and cytotoxic function of NK cells following cigarette smoke exposure [53]. The impairment of the immune system caused by traditional cigarette smoking diminishes the smokers’ ability to fight invading pathogens and increases the likelihood of developing infections in the respiratory tract. 

E-cigs also increase the pulmonary signaling for neutrophils and alveolar macrophages [54,55]. Following e-cig use, alveolar macrophages showed an increased production of pro-inflammatory molecules regardless of whether the e-cigarette liquid contained nicotine [54], Figure 1.

In some cases, cellular damage from e-cigs was found to be more extensive than the damage imposed by traditional cigarettes [56]. Comparing the exposure of different bronchial epithelial cell isolates from COPD patients with vaporized JUUL and a reference standard cigarette showed augmented cell cytotoxicity, especially in response to flavored e-cig aerosols and LDH secretion as previously reported [57]. A 2019 study on the airway smooth muscle cells (ASMCs) of COPD patients showed increased secretion of IL8 following e-cigarette exposure, supporting other reports that e-cigarettes lead to increased neutrophilic inflammation in pulmonary tissues [55]. The findings in this study suggest the progression of COPD may be accelerated by e-cigarette use [55]. The pathogenesis of COPD is highly associated with increased inflammation, particularly from macrophages and neutrophils [55]. Of note, as previously mentioned, even the carrier liquid component propylene glycol can negatively affect the airways: Using bronchial epithelial cells and an in vivo sheep model, it was demonstrated that exposure to e-cig aerosol of 100% propylene glycol resulted in increased mucus concentration as relevant to lung disease and COPD [58], Figure 1. The exposure of human airway epithelial cells to e-cig aerosols can increase the secretion of inflammatory cytokines, such as IL6 and IL8. Furthermore, human lung fibroblasts show stress responses and morphological changes upon e-cig exposure, such as an increased secretion of IL8 or even cell death, especially after treatment with cinnamon-flavored e-liquid [59].

Similarly, flavoring chemicals stimulated naïve THP-1 macrophages to produce significantly elevated levels of the pro-inflammatory cytokines IL1B, IL8, and TNFα when exposed to ethyl maltol, while other flavorings suppressed inflammatory cytokine secretion [60], Figure 1. In humans, a longitudinal cohort pilot conducted by Sayed et al. explored changes in the inflammatory state and monocyte function of e-cig users versus healthy controls and demonstrated an altered inflammatory state of the airways and systemic circulation, raising concern for the development of both inflammatory and infectious diseases in chronic users of e-cigs [61].

NF-kB has been shown to mediate acute lung inflammation in mice in a dose-dependent manner, with and without nicotine present in the e-cig aerosols [62]. Using human airway epithelial cells, Song et al. showed that the activation of the NF-kB and MAPK/ERK signaling pathways by e-cig aerosols can induce the expression of mucins, i.e., MUC5AC, as relevant to lung disease [63]. Our own study supports the activation of ERK and NF-kB signaling in oral epithelial cells [49].

As we will discuss in the following sections, e-cig use can increase the risk of lung disease by inducing oxidative stress and inflammation [64]. Vaping has also been linked to dysregulation in mitochondria, the key organelle of reactive oxygen production [65]. The link to mitochondrial dysregulation has further been supported by another study demonstrating that e-cig vape induces the upregulation of Toll-like receptor 9 (TLR9), which is associated with mitochondrial DNA damage, together elevating the risk for atherosclerosis [66], (see next section and Figure 2). Additionally, inflammation is associated with DNA damage, which, in constantly regenerating cells, contributes to an increased risk of mutagenesis and malignant transformation [38,67]. These concepts are further discussed in Section 2.3 and Section 2.4.

### 2.3. The Induction of Reactive Oxygen Species and Related Pathways by E-Cig Vape

Oxidative stress occurs when there is an imbalance between the production of reactive oxygen species (ROS) and antioxidants [68]. In moderation, ROS benefits the cell by regulating several cellular mechanisms that are protective against carcinogenesis. Specifically, ROS modulates antioxidant production, DNA repair, inflammatory responses, and cell growth and death. When the amount of ROS in the cell becomes excessive, the result is oxidative stress [68]. Oxidative stress may be caused by external cellular damage or by a failure of DNA repair systems [69]. The protein NRF2 (also known as NFE2L2) is central to the regulation of antioxidant gene expression [70]. Under homeostatic conditions, NRF2 remains in the cytoplasm where it is bound to KEAP1. KEAP1, together with CUL3 and RBX1, form the core ubiquitin ligase 3 complex. When NRF2 is bound to the complex, it is degraded by the proteasome, which prevents it from accumulating in the cytoplasm, Figure 2. When ROS levels rise, the binding of NRF2 to KEAP1 is disrupted. This allows NRF2 to escape protein degradation and to enter the nucleus. Once inside the nucleus, NRF2 can initiate antioxidant transcription by forming a heterodimer with MAF proteins and binding to target gene sites. The NRF2 signaling pathway regulates the transcription of over 500 genes and is an important mechanism of protection against oxidative stress [70], Figure 2.

Aerosols produced during a vape session contain oxidants and ROS that are generated during the heating of the liquid by the coil. The source of those oxidants was identified using a modified 2′-7′-dichlorodihydrofluorescein diacetate fluorescein derived dye that detects oxidant reactivity directly in e-cig liquids and aerosols. Even unvaporized e-liquids can have oxidative effects; however, flavor additives such as sweet and fruity flavors had the strongest oxidizer function [59]. Dripping, as previously described, also results in higher oxidant levels and reactive oxygen species release. The same group assessed the oxidant effect of e-cig aerosols in a mouse model and identified diminished lung glutathione levels in response to e-cig aerosol exposure. Glutathione (see Figure 2) is a key molecule of the cellular redox balance by participating in the removal of free radicals and ROS. It has been previously shown that mouse exposure to traditional cigarettes reduced glutathione levels in the lung [71]. The modification of glutathione levels in lung cells upon inhalation of e-cig aerosols most likely results in oxidative stress culminating in inflammatory responses as seen with conventional cigarette smoke [72,73], Figure 2.

It has also been reported that heavy metal particles in e-cig vape, i.e., copper nanoparticles, can alter mitochondrial reactive oxygen species (mtROS) and the stability of the electron transport chain (ETC) complex resulting in mitochondrial DNA damage [74]. Mitochondria responded acutely to direct e-cig aerosol exposure by increasing mtROS to higher relative levels compared to air-exposed cells. Complexes I and III are known to be the major sources of superoxide (O_2_^−^) mtROS. The authors measured a significant reduction in the stability of complex IV cytochrome C oxidase subunit II and proposed that this may result in inefficient transfer of electrons and possibly an “electron leak” leading to the formation of mtROS [74], Figure 2. Upon aerosol exposure, NAD(P)H quinone dehydrogenase (Nqo1), a protective antioxidant, was observed to be induced. The e-cig aerosol-mediated upregulation of Nqo1 suggests activation of the NRF2 pathway [74]. NRF2, which binds the antioxidant response element (ARE), see Figure 2, has so far mostly been shown to be induced by traditional smoking, not by e-cig vape [75]. If functional, the NRF pathway is highly important in mitigating potential carcinogenesis. However, it has been reported that common flavoring agents, including cinnamaldehyde, guaiacol, and eugenol, significantly activate the NRF2 pathway when compared to the media-only control in lung epithelial cells [76] attempting an antioxidative response. Acrolein inhalation in mice induced carbonyl deposition and the expression of KEAP1 in the lungs, interfering with NRF2 antioxidative function, Figure 2. In the vascular system, e-cig aerosols induce oxidative stress via NOX2, a phagocytic NADPH oxidase [77]. DNA damage (see Section 2.4) has been shown to increase intracellular levels of ROS and regulate cellular death via the overexpression of H2AX in a mechanism involving NOX1 and RAC1 [78], Figure 2. 

Smoking either electronic or traditional cigarettes can affect ROS production and increase the likelihood of developing COPD. The oxidation of polycyclic aromatic hydrocarbons (PAHs) and other harmful and potentially harmful constituents found in cigarette smoke and e-cigarette vapor increases ROS production [79]. In some instances, e-cigarette and traditional cigarette exposure stimulates an increase in ROS production in alveolar macrophages, which leads to pulmonary epithelial damage and an influx of neutrophils to the damage site [54]. The result of this is increased inflammation, mucus production, and destruction of alveolar cells causing airway obstruction [70]. 

### 2.4. E-Cig Vape Causes DNA Damage

Tobacco products and e-cig vapes contain compounds that can be enzymatically activated into toxic substances, which become mutagenic. The required enzymes function in two phases [80,81]. Phase I enzymes introduce an oxygen atom or reveal functional groups on the target metabolite. Cytochrome P450 is involved in the Phase I activation of many chemical compounds [80,81]. Chemicals in cigarette smoke, such as polycyclic aromatic hydrocarbons (PAHs), upon hydroxylation, can generate high intracellular levels of ROS and free radicals, which interact with DNA by forming oxidized lesions, having a genotoxic effect [80,81]. DNA adducts are a central mechanism of tumorigenesis by genotoxic agents. Nicotine-derived chemical carcinogens such as 4-(methylnitrosamino)-1-(3-pyridyl)-1-butanone (NNK) form adducts on the phosphate group of DNA [82]. NNK has been shown to thereby induce a transversion mutation (Figure 3) on the K-ras gene in pulmonary tissues [83]. PAHs may also cause a transversion mutation in the p53 gene, which is also strongly correlated with the development of lung cancer [84]. 

Similarly, acrolein readily reacts with DNA to produce well-characterized adducts such as (8R/S)-3-(2′-deoxyribos-1′-yl)-5,6,7,8-tetrahydro-8-hydroxypyrimido[1,2-a]purine-10(3H)-one or short γ-OH-Acr-dGuo. These were tested for in the oral cells of e-cigarette users and non-users [83]. γ-OH-Acr-dGuo DNA adduct formation was detected in the buccal mucosa and oral cavities of e-cigarette users compared to non-users, Figure 3. Unrepaired DNA adducts may cause cellular mutation by interfering with the normal replication and transcription cycles [82], Figure 3. DNA adduct formation at specific oncogene sites is, therefore, strongly correlated with carcinogenesis.

DNA adduct formation caused by the by-products of heating e-cigarette liquid can also contribute to the development of cancer because their presence causes mutations [23]. Overall, smoking and vaping damage DNA via DNA adduct formation and thereby increase the likelihood of carcinogenesis. DNA adducts are the product of covalent bonds of metabolically active carcinogens to DNA [82]. DNA damage is normally repaired by DNA polymerases. However, in cases where a large amount of DNA damage is sustained, repair mechanisms cannot meet cellular demands, and mutational events can manifest, leading to cancer [23]. E-cigarette aerosols also induced single- and double-DNA-strand breaks, Figure 3. Both nicotine-containing and nicotine-free aerosols increase DNA breaks compared to controls; however, nicotine-containing aerosols show greater genotoxicity. These findings were associated with altered cell cycle control and increased apoptosis [85]. The chronic use of e-cigs could result in repeated DNA damage. As DNA damage repair is error-prone, especially non-homologous end joining, it can allow for the accumulation of genomic aberrations.

Tobacco smoking roughly triples the accumulation of mutations per cell per year compared to that found in non-smokers [86]. Although e-cig vape is known to induce DNA damage [87,88], its effect on the mutation rate in lung or oral cells is unknown. Since e-cigs have only emerged within the last 15 years, the long-term effects of vaping on the mutational burden in vivo in humans have not been established. However, both vapers and smokers demonstrate significantly higher levels of DNA damage in their oral cells as compared to non-users. The levels of DNA damage increased dose-dependently in both vapers and smokers as compared to non-users. When focusing on e-cig users, those who used sweet-, mint or menthol-, and fruit-flavored e-liquids were detected to have the highest levels of DNA damage. Nicotine content was not a predictor of DNA damage in e-cig users [89]. Interestingly, in a study on vaping comprising human participants, significant mRNA expression changes were detected after vaping [23]. These changes included DNA damage and expression of repair genes, suggesting significant and acute DNA damage just with vaping 20 puffs over a period of 20 min.

Volatile organic compounds such as aldehydes induce DNA damage, and more so impair DNA repair mechanisms and OGG1 excision, and reduce the activity of damage detection by XPC [90], Figure 3. This is of importance, as aldehydes thereby induce higher levels of genotoxicity than nitrosamines, ketones, and nitrosonornicotine, which only depend on the activation of the cytochrome (CYP) pathway (see earlier in this paragraph). Aldehydes inhibit CYP-mediated detoxification and further impair DNA damage repair pathways [90,91,92], Figure 3.

If DNA damage cannot be repaired, it poses a risk that leads to abnormal cell behavior, which can cause cell death via reactive oxygen species (ROS) and DNA damage response (DDR) signaling [78]. These mechanisms are protective and designed to prevent propagation of cellular mutations. The DDR pathway halts the cell cycle at DNA damage sites until the damage is either repaired or the cell is subjected to apoptosis [78,93]. We have shown in our research that e-cig aerosol exposure of oral epithelial cells induces a dose-dependent cell viability reduction, regardless of nicotine content, in a possible attempt to repair DNA damage, as measured via pH2AX [46]. pH2AX foci mark the site of double-strand DNA breaks and recruit additional proteins involved in DDR and repair. At the completion of DNA repair, pH2AX is typically de-activated [94,95]. If it remains activated, cells may continue to replicate without proper DNA repair [95]. As relevant to tumorigenesis, pH2AX has been shown to be higher in dysplastic oral tissues and is significantly associated with disease progression to oral cancer [96].

### 2.5. The Increased Risk for Tumorigenesis upon E-Cig Aerosol Exposure

Cancer is associated with the cumulative acquisition of genetic defects. Mechanisms capable of inducing chronic inflammation and DNA damage are, therefore, involved in tumorigenesis. The smoking of traditional tobacco products or e-cigarette vaping have been described to cause inflammation and DNA mutations: As mentioned in Section 2.4, nicotine can be metabolized to the carcinogenic nitrosamine ketone (NNK) and N′-nitrosonornicotine (NNN). E-cigarette usage may predispose users to the development of cancers just as traditional cigarette smoking does [34]. NNN is present in the saliva of e-cigarette users, in many cases showing similar NNN concentrations to those found in the saliva of traditional cigarette smokers [34]. Exposure to NNN can result directly from nicotine consumption, or it can form via the nitrosation of nicotine and nornicotine. Reports indicate that NNN production likely occurs in the oral cavity, where nornicotine can interact with nitrite. Other studies using rat models showed that NNN exposure resulted in the formation of tumors in the esophagus and the oral mucosa [34].

The most direct evidence to date linking vaping and cancer comes from an in vitro study demonstrating that flavored and unflavored e-cig aerosols transformed bronchial epithelial cells [97]. Additionally, Lee et al., like others, found that e-cig vape results in DNA damage and reduced repair in murine lung, heart, and bladder tissue [88], but, more concerningly, that this subsequently caused lung adenocarcinoma and bladder hyperplasia [98].

In 2017, a case report was released linking chronic e-cig usage to the development of oral cancer in two different male patients [99]. In 2021, a 19-year-old heavy e-cig user presented with an oral ulcer, which developed into an aggressive poorly responsive fatal squamous cell carcinoma [100]. No other carcinogenic risk factors such as HPV were identified, suggesting a causative role of e-cig vaping in this case.

While we discussed the molecular and cellular alterations in response to e-cig vape exposure in detail, a brief description of the resulting tissue damage is to follow: E-cig aerosols penetrate deep into the lung tissue and can affect the healthy lung in never-smokers, even after only short e-cig vape inhalation [101]. E-cig users report dryness of their throat, cough, and soreness [102,103]. Oral mucosal symptoms, including black tongue, burns, nicotine stomatitis (smoker’s palate), and hairy tongue, have been shown to be significantly more common in e-cig users than in ex-smokers. While milder and more transient than in tobacco smokers, the symptoms and their intensity were reported to be related to the flavors used, e.g., menthol and cinnamon, which can result in oral irritation. Other flavors such as citrus, sour, and custard flavors are also associated with severe throat symptoms [104]. Interestingly, hairy tongue and nicotine stomatitis were significantly more prevalent in consumers of e-cigarettes relative to former smokers [105]. Additionally, tooth loss and cheek pain have been reported by e-cig users [106].

Given the short amount of time that e-cigarettes have been on the market, the longer-term safety of these products remains unknown. Considering that cancer takes decades to develop, the findings of malignant precursors, tumor-initiating gene expression changes, and DNA-damaging events associated with mutation bode a potentially dark future for e-cig users.

## 3. Conclusions

The long-term consequences of e-cigarette vaping will not be known for another decade. However, in recent years, it has been shown that vaping is not a safe alternative to traditional smoking. We summarized a comparison in Table 1. Youth have been targeted with sweet and fruity flavors and the sleek design of the e-cig devices. It is important to note that the alluring flavors and e-cig liquid alone are toxic, even in the absence of nicotine. The health risks associated with vaping affect multiple organs: the oral cavity, lungs, vascular system, and brain. While there is currently no direct evidence that e-cig use increases cancer risk in humans, it is well recognized that e-cigs initiate precursor events to cancer, such as inflammation and DNA damage. Tumor models further demonstrated that DNA damage and DNA repair inhibition induced by e-cig aerosols resulted in lung cancer and bladder precancer in rodent models. This implies that the long-term health implications for a new generation of young people are grave, including addiction to nicotine and an increased risk for preventable cancers.

## Figures and Tables

**Figure 1 cells-12-02552-f001:**
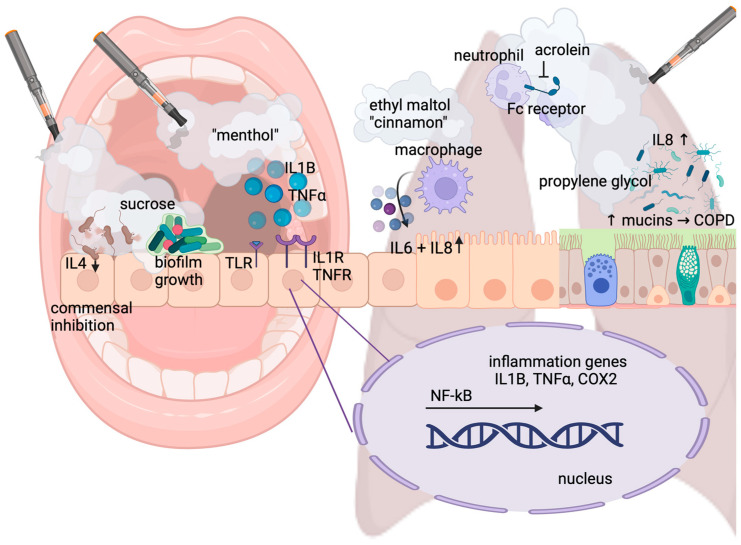
Pathways of inflammatory signaling. E-cig aerosols induce the expression of pattern-recognition receptors such as TLR and the secretion of pro-inflammatory cytokines such as IL1B and TNFα and their respective receptors, particularly in response to menthol-flavored e-cigs. In the oral cavity (left part of the figure), e-cig vape affects the healthy oral microbiota by inhibiting commensals and favoring growth and biofilm formation of cariogenic bacteria, especially in the presence of sweet flavors containing sugar. IL4 suppression further supports colonization with pathogenic bacteria. In human airway epithelial cells (right part of the figure), e-cig aerosols also increase inflammatory cytokines, such as IL6 and IL8, secreted by macrophages in response to, for example, cinnamon- and caramel-flavored e-vape containing ethyl maltol. Acrolein, found in e-cig vape, inhibits the Fc receptor and function of neutrophils. Additionally, exposure to even a 100% propylene glycol e-cig liquid without flavors and nicotine results in increased mucus concentration, trapping bacteria and viruses and leading to IL8-mediated inflammation relevant to lung disease and COPD.

**Figure 2 cells-12-02552-f002:**
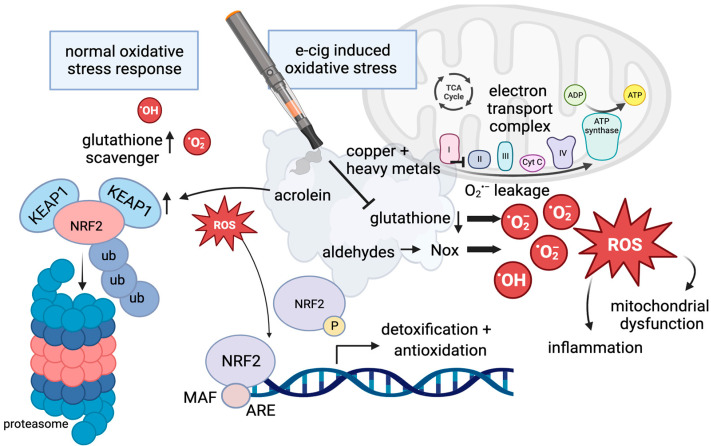
The regulation of reactive oxygen species. Under normal conditions, glutathione functions in scavenging reactive oxygen species, and levels of NRF2 remain low due to ubiquitin-mediated degradation in the proteasome (left panel). If oxidative stress increases, NRF2 escapes protein degradation upon phosphorylation and separation from KEAP1, and it enters the nucleus to regulate gene expression by binding the antioxidant response element (ARE) as a normal reaction in response to oxidative stress. Flavoring, acrolein, and aldehydes can interfere with the normal oxidative stress response: Acrolein can induce the expression of KEAP1, and aldehydes induce NOX proteins, which can result in high levels of ROS. Copper nanoparticles in e-cig aerosols increase mitochondrial dysfunction, as a consequence of mtROS associated with an electron leak. In the presence of e-cigs, glutathione levels are diminished and ROS created by e-cig exposure, including unvaporized e-liquids and flavorings, cannot be scavenged.

**Figure 3 cells-12-02552-f003:**
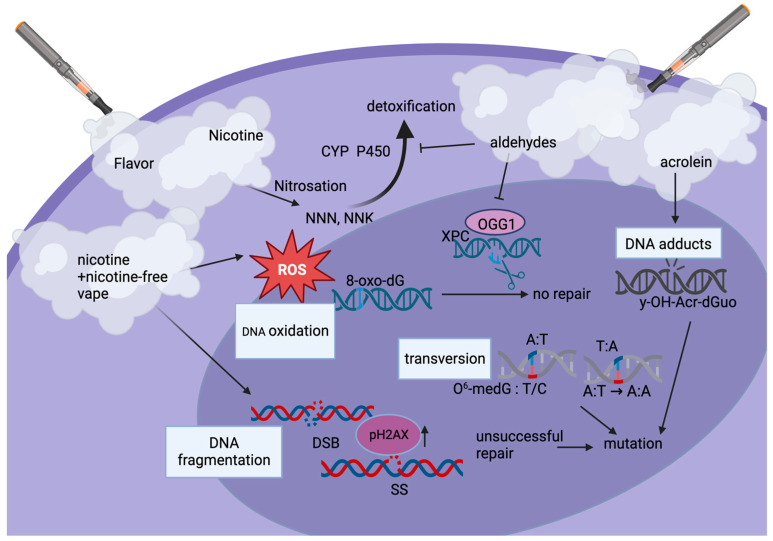
Types of e-cig-induced DNA damage. The dominant pathway of nicotine metabolism in humans is the formation of cotinine, the first step of which is catalyzed by cytochrome P450 (CYP P450). Nicotine can be metabolized to 4-(methylnitrosamino)-1-(3-pyridyl)-1-butanone (NNK) and N’-nitrosonornicotine (NNN). Both of these compounds, as well as acrolein (y-OH-Acr-dGUO), can lead to DNA adduct formation. E-cig aerosol-induced reactive oxygen species (ROS) can also induce DNA oxidation, such as 8-oxo-dG. Usually detected by XPC and repaired via excision by OGG1, aldehydes in the e-cig vape inhibit this repair step causing potential mutations. Nicotine and nicotine-free vape induce DNA fragmentation, including DNA single (SS)- and double-strand breaks (DSB), which require DNA damage repair proteins such as pH2AX to be recruited. Transversions (base-pair substitutions) lead to mutations, e.g., in genes such as Ras and p53. Aldehydes suppress DNA damage repair as they inhibit cytochrome (CYP)-mediated detoxification. Overall, levels of DNA damage correlate with the amount of vape consumed as well as additives such as flavoring, especially sweet, fruit, and menthol flavors.

**Table 1 cells-12-02552-t001:** A comparison of mechanisms of traditional and e-cigarette-induced cell damage and outcome.

	**Traditional Cigarette**	**E-Cigarette**
Immune response and associated inflammatory diseases	Increased inflammation and DNA damage [107]	Increased inflammation and DNA damage [45]
	Oral tissues: Periodontitis and pathogenic bacterial growth [41]	Oral tissues: Periodontitis and pathogenic bacterial growth [43,44,45,49]
	Respiratory ailments: COPD and Asthma [51] Infection [52] Impaired pulmonary immune response → lung damage [53]	Respiratory ailments: COPD [55,56,57,58] Infection [61] Impaired pulmonary immune response → lung damage [53,54,55]
Inflammatory signaling mediators	ERK1/2 ↑	MAP2K6 [23] ↑
	AP-1 ↑	AP-1 not known
	Sox2 [108] ↑	Sox2 [108] ↑
	JAK-2/STAT-3 [109] ↑	Not reported but ROS can activate JAK/STAT and Toll signaling pathways
	AMPKa2 [110] ↑	AMPKa2 [110] ↑
	EGFR [111] ↑	EGFR [111] ↑
	Wnt [112,113] ↑	Wnt [112] ↓
	COX2 [114,115]	COX2 [45,49,50] ↑
ROS-related damage and its clinical implications	Glutathione is reduced in the lung [71]	Glutathione is reduced in the lung [72,73]
	NRF2 is activated [75]	NRF2 seems not to be directly activated by e-cig vape but flavoring agents in e-cig vape activate NRF2 [76]
	Mitochondrial damage: Fragmentation and disruption of the mitochondrial network caused by increased expression of Drp1 and decreased Mfn 2. Increased risk for various respiratory diseases [116,117,118]	Mitochondrial damage: Dysregulation due to mtDNA damage induced by TLR9 ↑ → risk for atherosclerosis [65,66]
	Oxidative stress (ROS) → lung disease [54,79]	Oxidative stress (ROS) → lung disease [54,64,79]
Mechanisms by which chemicals contribute to ROS and DNA damage	** Nicotine (Nitrosamine) ** ROS DNA damage Alkyl phosphotriester formation [23]	** Nicotine (Nitrosamine) ** ROS DNA damage Alkyl phosphotriester formation [23]
	** Acrolein ** Neutrophil inhibition ROS	** Acrolein ** Neutrophil inhibition ROS DNA adducts
	** Aldehyde ** DNA adducts [90]	**Aldehyde** ROS [16,17] DNA adducts [90]
	**Nanoparticulate carbon black (nCB) ** [119] Apoptosis of phagocytic cells Increased inflammatory signaling	**Nanoparticulate carbon black (nCB)** N/A
	**Polycyclic aromatic hydrocarbons ** [15] ROS DNA adducts	**Polycyclic aromatic hydrocarbons ** [15] ROS DNA adducts
	** Copper and Heavy Metals ** Copper detected in saliva of individuals with environmental tobacco exposure [120] Other toxic heavy metals identified in cigarettes [121]	**Copper and Heavy Metals** [74] Mitochondrial ROS DNA damage Increased inflammatory signaling
	**Propylene glycol/Vegetable glycerol ** N/A	**Propylene glycol/Vegetable glycerol ** [12,23] Carcinogenic byproduct formation DNA adducts
	**Diacetyl** N/A	**Diacetyl** [25] DNA damage Linked to lung damage and cancer [24]
Tumorigenesis Risk and Its Relationship to Known Cancers	Accumulation of mutations per cell per year triples compared to non-smokers [86]	Mutational load unknown, but evidence of DNA damage and inhibited repair [82,90,91,92]
	Carcinogenic NNN levels high [34]	Carcinogenic NNN levels similar to traditional smoking [34]
	Main risk factor for human lung cancer [122]	Lung cancer in animal model [98]
	Oral cancer in tobacco and smokeless tobacco users [123]	Oral cancer in heavy users [99,100]

## Data Availability

Data sharing is not applicable to this article.

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
