# Peer review of "Mechanisms of E-Cigarette Vape-Induced Epithelial Cell Damage"

_cells, 2023, doi:10.3390/cells12212552_

Round 1
Reviewer 1 Report
Comments and Suggestions for Authors
Hello,
The review paper titled “Mechanisms of E-cigarette vape-induced epithelial cell damage” explores the potential cellular effect on the airway linings by vaping. Though it considered safe and acceptable approach as vape compared to smoke, there are unavoidable consequences that one needs to accept mainly on epithelial and ASM cells structure.
Authors have summarized the available literature well and cellular degeneration figure for their mechanistic effect is good addition. Overall, the review paper is in good shape, and it would definitely open scope for further research in many domains.
Thank you,
Author Response
We would like to thank the reviewers for the positive comments on the submitted manuscript.
Reviewer 2 Report
Comments and Suggestions for Authors
Although the topic of the review is interesting, my initial enthusiasm was dampened when I read the paper and found out the title (and abstract) of the paper does not adequately reflect its content. Indeed, there could have been more emphasis on molecular mechanisms underlying e-cig-induced health effects. In fact, there is almost no information regarding these mechanisms in epithelial cells (as the title implies). Furthermore, I feel the figures do not add any relevant information to the manuscript as they describe basic mechanisms of inflammation/oxidative stress/mitochondrial dysfucntion that are already well-known and abundantly described. In conclusion, in my opinion the paper, in its current form, does not add enough significantly relevant new information to the field. The paper would have benefitted froma more detailed description of the impact of e-cig emissions (or components thereof) on signaling pathways underlying inflammation, ox stress, mitochondrial dysfunction (and its underlying pathways as mitochondrial biogenesis/mitophagy) etc in airway/lung epithelial cells. Using/synthesising this info into new figures to comprehensively summarize and visualize this info would strenghten the manuscript significantly. Furthermore, as much is known about the effects of (components) of cigarette smoke on these molecular mechanisms/pathways, it would benefit the manuscript if (at least short or summarized) the comparison with cigarette smoke vs e-cig emissions could have been made.
Author Response
We would like to thank the reviewer for the valuable critics of the manuscript in its current form and the suggestions for improvement.
Comment: “there is almost no information regarding these mechanisms in epithelial cells (as the title implies)”
Reply: We would like to clarify that most of the findings we summarized in this review were based on studies using epithelial cells of the lung and oral cavity. The only other cell type reportedly playing a role in inflammation we focused on were macrophages (now ref 37, 53-56, 60).
Comment: “The paper would have benefitted from a more detailed description of the impact of e-cig emissions (or components thereof) on signaling pathways underlying inflammation, ox stress, mitochondrial dysfunction (and its underlying pathways as mitochondrial biogenesis/mitophagy) etc. in airway/lung epithelial cells.”
Reply: We like to thank the reviewer for drawing attention to the importance of the components. We highlighted in the text and more the modified figures which component, e.g., acrolein, aldehyde and others cause epithelial damage.
Comment: “the figures do not add any relevant information to the manuscript as they describe basic mechanisms of inflammation/oxidative stress/mitochondrial dysfunction that are already well-known and abundantly described. Using/synthesizing this info into new figures to comprehensively summarize and visualize this info would strengthen the manuscript significantly.”
Reply: We like to thank the reviewer for commenting on the basic nature of the figures. We agree that the figures only provided an overview over known mechanisms, and we modified all three Figures.
Comment: “Furthermore, as much is known about the effects of (components) of cigarette smoke on these molecular mechanisms/pathways, it would benefit the manuscript if (at least short or summarized) the comparison with cigarette smoke vs e-cig emissions could have been made.”
Reply: While our goals was to provide a comparison between cigarette smoke and e-cig vaping, it seems we needed to highlight which findings were from cigarette We therefore added “traditional” to all cigarette vs e-cig comparisons.
Reviewer 3 Report
Comments and Suggestions for Authors
This review emphasizes that e-cig may not be a safe alternative to traditional cigarettes. This is a critical conclusion that could provide valuable insights for diseases (particularly cancer) prevention and treatment, and policy-making.
However, to make the logical flow clearer, the authors need to reorganize the paragraphs, and revise the figures.
The first section is too long, which should only serve as a background. I suggest the authors focusing on the e-cig, answer the questions what is the e-cigs? What is the advantage of e-cig over traditional cigar or cigarette? What are the potential health concerns of e-cig? And I think the second section can be combined into the first section.
For the second section, as I mentioned above, the content about the e-cig use in youth and adolescents can be moved to the first section. Because this content is only discussed in the first two paragraphs and part of the 5th paragraph. In other parts of this section, the authors are talking about the risk of e-cig to common people, which should be combined to the later sections (or the first section, as a background introducing).
Figure 1 is not clear. As the authors referred Figure 1 in line 169, to show the PRRs (by the way, pattern-recognition receptor, not patter-recognition receptor) associated with inflammation, but this information is not clear labeled in Figure 1. In line 258, the authors claiming that macrophages in alveoli produced pro-inflammatory molecules after use e-cigs even without nicotine. But none of these information is clearly shown in Figure 1 except for a macrophage cell. This problem also apply to line 281 and line 334. Moreover, the infectious pathogen factors (such as bacteria and viruses), which were an important part of this section, were not shown in this figure.
Line 170 to 189, this part talked about the inflammation reaction to the injury. This part could be minimized because it was again background. And the authors should focus on how e-cigs inducing inflammation.
Line 196 to 199, this claim is very interesting. But with some citation of the numbers, it will be more convincing. e.g., "significantly worse", how worse? Is there a percentage? And I think it would be better if the authors could provide any explanations of this difference between e-cig users and conventional smokers.
Line 214 to 217. Are there any evidences that e-cigs produce sugars and/or lactic acid in oral environment? Please add one sentence stating the relationship between e-cig and sugars/lactic acid.
Line 228, DNA damage was reviewed in section 3.3 instead of 3.4.
Line 229 to 254, this whole paragraph talked about the traditional cigarettes, which is a distraction to the main theme: e-cig. I understand that the research on cigarettes are much more than e-cig. However, I still suggest the authors focusing on e-cig. They should provide more information about the cellular mechanism of e-cig inducing pulmonary inflammation (by extending Line 255 to 258). Then briefly review some cigarette research as comparison.
Line 259 to 260, this claim also can be explained in details.
Figure 2 is better than Figure 1. But there are still some parts confusing. What do the "normal" and "scavenging" next to the e-cig mean? What is the arrowhead pointing to the Complex III? Why there are two NRF2-p?
Line 361 to 364, how does Figure 2 support this sentence?
Line 415 to 417, the information in this sentence is not included in Figure 3.
Line 449 to 462 look like a duplication of section 3.3. I understand that DNA damage can lead to cancer. But the authors should still reorganize the paragraphs so that the readers can get new information more easily.
Author Response
Comment: This review emphasizes that e-cig may not be a safe alternative to traditional cigarettes. This is a critical conclusion that could provide valuable insights for diseases (particularly cancer) prevention and treatment, and policy-making.
Reply: We thank the reviewer for recognizing the importance of raising awareness about the risk associated with e-cigarette vaping.
We also appreciate the detailed instructions on how to improve the manuscript and organizing the information better.
Comments: The first section is too long, which should only serve as a background. I suggest the authors focusing on the e-cig, answer the questions what is the e-cigs? What is the advantage of e-cig over traditional cigar or cigarette? What are the potential health concerns of e-cig? And I think the second section can be combined into the first section.
Reply: We followed the reviewer’s suggestions to streamline the information and to highlighten the most important findings, and deleted paragraphs not essential to the review topic.
Comments on Figures (combined):
Figure 1 is not clear. As the authors referred Figure 1 in line 169, to show the PRRs (by the way, pattern-recognition receptor, not patter-recognition receptor) associated with inflammation, but this information is not clear labeled in Figure 1. In line 258, the authors claiming that macrophages in alveoli produced pro-inflammatory molecules after use e-cigs even without nicotine. But none of these information is clearly shown in Figure 1 except for a macrophage cell. This problem also apply to line 281 and line 334. Moreover, the infectious pathogen factors (such as bacteria and viruses), which were an important part of this section, were not shown in this figure.
Figure 2 is better than Figure 1. But there are still some parts confusing. What do the "normal" and "scavenging" next to the e-cig mean? What is the arrowhead pointing to the Complex III? Why there are two NRF2-p?
Line 361 to 364, how does Figure 2 support this sentence?
Line 415 to 417, the information in this sentence is not included in Figure 3.
Reply: All figures have been modified to represent mechanistic findings described in the text.
Comment: Line 196 to 199, this claim is very interesting. But with some citation of the numbers, it will be more convincing. e.g., "significantly worse", how worse? Is there a percentage? And I think it would be better if the authors could provide any explanations of this difference between e-cig users and conventional smokers.
Reply: We added the millimeter measurements of clinical attachment loss for a better understanding of quantification.
Comment: Line 214 to 217. Are there any evidences that e-cigs produce sugars and/or lactic acid in oral environment? Please add one sentence stating the relationship between e-cig and sugars/lactic acid.
Reply: We expanded that sentence to include a new reference detailing which sugars are present in sweet e-cig flavors: Fagan P, Pokhrel P, Herzog TA, Moolchan ET, Cassel KD, Franke AA, Li X, Pagano I, Trinidad DR, Sakuma KK, Sterling K, Jorgensen D, Lynch T, Kawamoto C, Guy MC, Lagua I, Hanes S, Alexander LA, Clanton MS, Graham-Tutt C, Eissenberg T; Addictive Carcinogens Workgroup. Sugar and Aldehyde Content in Flavored Electronic Cigarette Liquids. Nicotine Tob Res. 2018 Jul 9;20(8):985-992. doi: 10.1093/ntr/ntx234. PMID: 29182761; PMCID: PMC6037055.
Comment: Line 228, DNA damage was reviewed in section 3.3 instead of 3.4.
Reply: Thank you! Fixed.
Comments: Line 229 to 254, this whole paragraph talked about the traditional cigarettes, which is a distraction to the main theme: e-cig. I understand that the research on cigarettes are much more than e-cig. However, I still suggest the authors focusing on e-cig. They should provide more information about the cellular mechanism of e-cig inducing pulmonary inflammation (by extending Line 255 to 258). Then briefly review some cigarette research as comparison.
The other reviewer suggested to provide more information on traditional smoking, so decided to refer to both but emphasize the findings from traditional smoking vs. E-cig.
Comments: Line 259 to 260, this claim also can be explained in details.
Reply: We added more detail and a new reference to this sentence. Ween, M.P., Hamon, R., Macowan, M.G., Thredgold, L., Reynolds, P.N. and Hodge, S.J., 2020. Effects of E‐cigarette E‐liquid components on bronchial epithelial cells: Demonstration of dysfunctional efferocytosis. Respirology, 25(6), pp.620-628.
Comment: Line 449 to 462 look like a duplication of section 3.3. I understand that DNA damage can lead to cancer. But the authors should still reorganize the paragraphs so that the readers can get new information more easily.
Reply: This section has been deleted in the cancer section and moved to DNA damage.
Round 2
Reviewer 2 Report
Comments and Suggestions for Authors
Although the authors revised the manuscript, I still have some concerns and comments.
Major comments:
1) The section regarding E-cig modulated inflammatory signaling needs revision. More specifically: please add relevant literature regarding the impact of E-cigs on inflammatory signaling pathways (include the most well-known like NF-kB, AP-1 etc). Also include the impact of traditional cigarette smoke (and its components as aldehydes for example) in this section. For the latter there is abundant literature that you can use. This comparison of the effects of E-cig vs traditional cigs on these pathways will also significantly enhance the novelty of the paper.
2) The section on inflammatory signaling would really benefit from making the distinction between data obtained in different cell types better. For example using subheadings for each cell type/body compartment would already improve the structure of this section. This strategy can be applied to all other sections in the manuscript. Along the same line: the comparison between impact of E-cig or tradtional cig on specific processes in the review is not well-structured and difficult to assess. Simply adding the word ''traditional'' did not improve this. Please re-organize the text or use subheadings for cigarettes and E-cigs. Either way, the comparison between the impact of these 2 products should be more clear in the structure of the whole manuscript.
3) Please include a comparative (E-cig vs traditional cigs) description of the impact of these products on the molecular mechanisms controlling mitochondrial function (mitochondrial biogenesis vs mitophagy). There is a lot of evidence for deregulation of these pathways in response to cigarettes. Whether or not E-cigs modulate these pathways is relatively less described. If there is not enough evidence for this on E-cigs themselfs, the authors can speculate based on literature describing the impact of constituents of E-cig smoke on these pathways (for example aldehydes). This will also significantly enhance the novelty and quality of the paper.
Minor comments:
1) There are multiple typo's: TNFa (should be alpha), NFkbeta (should be NF-kB), IL1b (should be beta). Also sometimes the greek symbols are even missing and just TNF is stated (for example). Please correct and be consistent.
2) Since revision, the numbering of the paragraphs is not correct anymore. For example: there is no heading 2, there ar 2 sections 3.1...Please check and change accordingly
Comments on the Quality of English Language
English language is ok
Author Response
We'd like to thank the reviewer for the careful reading of the revised manuscript. We replaced the greek letters for consistency and corrected the numbering of the sections accordingly.
We also added a new paragraph and references to address the importance of NF-kB signaling (line 263-268). Currently, there are no publications in PubMed reporting the regulation of AP-1 in regard to e-cigarette vape, and as the reviewer pointed out little is known about mitochondrial function in the context of e-cigs. As we have no expertise in mitochondrial biogenesis vs mitophagy, we refrained from speculating on the impact of e-cig vapes.
As pointed out before, the request was made to focus on e-cig vaping and to not include cigarette smoking, so we did not include further information in aldehydes and in depth comparisons.
Reviewer 3 Report
Comments and Suggestions for Authors
Thank you for your modifications. Now the logic flow seems much better. All my concerns are addressed.
Author Response
Thank you!
Round 3
Reviewer 2 Report
Comments and Suggestions for Authors
After seeing the response of the authors on my requests, I still believe the novelty and quality of the manuscript is too limited. I would have really liked to see a comparison with conventional cigarettes and speculation about potential mechanistic involvement of specific chemicals in the detrimental effects of e-cigs as indicated in my comments. In my opinion this is necessary to allow publication in Cells.
As the authors are apparently not willing to include this, unfortunately in my opinion the paper (in its current form) is not suitable for publication in Cells.
Author Response
We already addressed the minor comments (typos and numbering) in the last version as well as included NF-KB. To address the reviewer's suggestion to compare traditional and e-cigarettes in more detail, we now added a table following the editor's guidance to highlight similarities and differences. This table also addressed briefly the damage on mitochondria.